# Comparative Performance of COVID-19 Test Methods in Healthcare Workers during the Omicron Wave

**DOI:** 10.3390/diagnostics14100986

**Published:** 2024-05-08

**Authors:** Emma C. Tornberg, Alexander Tomlinson, Nicholas T. T. Oshiro, Esraa Derfalie, Rabeka A. Ali, Marcel E. Curlin

**Affiliations:** Department of Medicine, Division of Infectious Diseases, Oregon Health and Sciences University, Portland, OR 97239, USAcurlin@ohsu.edu (M.E.C.)

**Keywords:** COVID-19, diagnostic testing, SARS-CoV-2, healthcare workers, PCR, rapid antigen, occupational health, signs and symptoms

## Abstract

The COVID-19 pandemic presents unique requirements for accessible, reliable testing, and many testing platforms and sampling techniques have been developed over the course of the pandemic. Not all test methods have been systematically compared to each other or a common gold standard, and the performance of tests developed in the early epidemic have not been consistently re-evaluated in the context of new variants. We conducted a repeated measures study with adult healthcare workers presenting for SARS-CoV-2 testing. Participants were tested using seven testing modalities. Test sensitivity was compared using any positive PCR test as the gold standard. A total of 325 individuals participated in the study. PCR tests were the most sensitive (saliva PCR 0.957 ± 0.048, nasopharyngeal PCR 0.877 ± 0.075, oropharyngeal PCR 0.849 ± 0.082). Standard nasal rapid antigen tests were less sensitive but roughly equivalent (BinaxNOW 0.613 ± 0.110, iHealth 0.627 ± 0.109). Oropharyngeal rapid antigen tests were the least sensitive (BinaxNOW 0.400 ± 0.111, iHealth brands 0.311 ± 0.105). PCR remains the most sensitive testing modality for the diagnosis of COVID-19 and saliva PCR is significantly more sensitive than oropharyngeal PCR and equivalent to nasopharyngeal PCR. Nasal AgRDTs are less sensitive than PCR but have benefits in convenience and accessibility. Saliva-based PCR testing is a viable alternative to traditional swab-based PCR testing for the diagnosis of COVID-19.

## 1. Introduction

The dramatic appearance of the highly transmissible SARS-CoV-2 pandemic on the world stage in late 2019 highlighted the importance of rapid, point-of-care diagnostic tests for respiratory illnesses. By 2022, a wide range of testing modalities had been developed for COVID-19, generally relying on either nucleic acid amplification (NAAT) or detection of viral antigens using lateral flow chromatography (antigen rapid diagnostic tests, or Ag-RDTs). NAAT-based tests are generally regarded as the most sensitive testing modality but require laboratory processing, have longer result turnaround times, and may yield false positive results in those with recently resolved infection. Use of at-home Ag-RDTs is widespread, with 20% of individuals reporting the use of one of these tests during the last 30 days of the period of omicron variant dominance (omicron wave) [1]. Pre-omicron data showed the sensitivity of a single Ag-RDT to be approximately 70%, with a positive correlation between higher viral load as estimated using cycle threshold (Ct) values and sensitivity [2,3,4]. Early studies during the omicron wave showed a range of sensitivities across Ag-RDT brands [5], with some, such as the BinaxNOW, demonstrating a consistent sensitivity of around 65% [6]. Other brands such as iHealth show similar sensitivity in detecting delta and omicron variants in the laboratory but have not been widely validated in epidemiologic studies during the delta and omicron waves [7]. 

Several factors may affect the differing performance of AgRDTs and NAATs over time. Many of these tests were developed and validated prior to the appearance of newly divergent strains such as the B.1.1.529 (omicron) variant of concern and subsequent viral strains [8]. The rapidly evolving nature of variants and subvariants of SARS-CoV-2 led to questions regarding the performance characteristics of diagnostic tests developed and validated prior to the appearance of contemporary strains, given that virus changes that may affect the magnitude and time course of viral RNA and antigen expression in various clinical samples. For NAATs, the most common of which use polymerase chain reaction (PCR) technology, a variety of sample sources have been evaluated including nasopharyngeal swabs (NP), oropharyngeal swabs (OP), and saliva samples. Pre-omicron meta-analyses showed either equivalence in saliva and NP testing [9], or slightly increased sensitivity of NP testing compared with saliva [10,11,12]. However, some studies involving early omicron strains showed better sensitivity with saliva swabs compared to mid-turbinate swabs [13,14]. NP testing is more costly than saliva and can be associated with significant discomfort [15]. AgRDTs are generally performed by swabbing the nares bilaterally [16,17], but the evidence of possibly better sensitivity in saliva PCR compared to NP PCR raises the question of whether OP AgRDT testing could provide equal or better sensitivity. Given the practical challenges associated with COVID-19 testing, including the multiplicity of platforms and testing sites, patient experience, and cost, as well as the broad relevance of these issues to the diagnosis of currently circulating respiratory viruses such as COVID-19, RSV, influenza and future respiratory epidemics, we performed a repeated measures observational study in a sample of adult healthcare workers presenting consecutively for SARS-CoV-2 testing at the Oregon Health and Sciences University Occupational Health (OHSU) clinic. 

## 2. Materials and Methods

Regulatory approval: This study was performed with informed consent from all participating individuals, with approval and regulatory oversight by the Oregon Health and Sciences University Institutional Review Board. 

Recruitment: We designed a repeated measures study evaluating PCR tests, iHealth Ag-RDTs (iHealth Labs, Inc.; Sunnyvale, CA, USA), and BinaxNOW Ag-RDTs (Abbott; Abbott Labs, IL, USA) at the OHSU Occupational Health Testing Site. Between 25 January 2022 and 4 March 2022, individuals presenting for SARS-CoV-2 testing for any reason at OHSU Occupational Health were approached consecutively and offered screening for enrollment on a first-come-first-serve basis. Individuals who had tested positive for COVID-19 in the last 90 days were excluded due to risk of false-positive results. Participants were required to be over the age of 18. Following consent, we evaluated each participant using 7 testing methods, as follows: PCR (nasopharyngeal swab, oropharyngeal swab, saliva sample); BinaxNOW Ag-RDT (nasal swab, oropharyngeal swab); and iHealth Ag-RDT (nasal swab, oropharyngeal swab). 

PCR testing: Researchers collected NP and OP swabs from participants in accordance with CDC guidelines [18]. Participants undergoing saliva testing were required not to eat or drink anything for at least 30 min prior to sample collection. They were then directed to spit pooled oral saliva into the collection tube until it reached the volume specified by the manufacturer (approx. 1 mL). Saliva samples were stored on ice and all samples were transported to the OHSU Molecular Microbiology Laboratory for testing on the day of collection. The viral RNA was extracted using the King Fisher MagMAX Viral/Pathogen Nucleic Acid Isolation Kit (Thermo Fisher Scientific Inc., Waltham, MA, USA). RNA was then reverse-transcribed and amplified using the Taqpath™ COVID-19 Multiplex RT-PCR (Multiplex) platform with software version v1.5.1 (Thermo Fisher Scientific Inc., Waltham, MA, USA), which targets the COVID-19 S-gene, N-gene, and ORF1ab. A positive test result required positive readings in 2/3 targets at a cycle threshold <40. Five samples were tested instead on the Hologic Panther (software version number V1.0.0; Hologic, Inc.; Bedford, MA, USA) or cobas^®^ 6800 (software version number 1.4; Roche Diagnostics; Indianapolis, IN, USA) systems due to logistical factors in the laboratory. Additionally, four samples were originally deemed inconclusive using Multiplex testing, and were re-tested on either the Hologic Panther or cobas^®^ 6800. These positive or negative results were included in sensitivity and specificity analysis. Only results from the Multiplex were used in Ct value analysis.

AgRDT testing: Researchers collected and tested nasal swabs in accordance with manufacturer specifications [16,17], except that samples were collected and tested outdoors at ambient temperatures ranging from 40 to 60 °F (the manufacturers suggest testing at “room temperature”). OP Ag-RDT samples were collected in accordance with CDC OP guidelines and samples were run otherwise in accordance with manufacturer guidelines. All samples were read out after the recommended run time by the collecting researcher. 

Statistical Analysis: Symptom data, test results, and sequencing data were described using frequencies and percentages. Odds ratios for symptoms and pairs of symptoms were calculated from contingency tables. Using a positive test on any PCR test as the gold standard, contingency tables were generated for each testing modality, and sensitivities and specificities were calculated. McNemar’s test was used for *p*-values. A *p*-value of <0.05 was considered statistically significant. Statistical analyses were conducted using Python version 3.10.8. 

## 3. Results

### 3.1. Patients and Samples

After several hundred participants were approached and screened, 325 individuals participated in this study; 6 were not tested with BinaxNOW kits due to supply shortages, 1 did not receive OP iHealth testing due to user error, and 32 were not tested in saliva due to eating or drinking within 30 min of the time of testing. All others were tested using all testing modalities. In total, 75 individuals were positive on at least one test; 8 individuals were positive on a single test; 19 individuals were positive on one or more PCR tests but negative on all antigen rapid detection tests; zero were negative on all PCR tests but positive on one or more AgDT; and 13 individuals were positive on all tests performed. The distribution of number of positive tests per individual and types of positive tests per number of positive test are shown in Figure 1. 

### 3.2. Symptoms

A total of 24 different symptoms were reported across all participants; 17 of these were reported in individuals that tested positive (Table 1). Sore throat (all participants = 50.7%, positive participants = 61.3%), cough (all participants = 26.8%, positive participants = 57.3%), and headache (all participants = 25.2%, postive participants = 32.0%) were the most commonly reported symptoms for both all and positive participants. Odds ratios (OR) for single symptoms ranged from 0.00 (nausea) to 6.29 (95% CI 3.58–11.03) (cough) (Table 2). Odds ratios of a positive test were highest for the combinations of chills and cough (29.73 95% CI 3.65–241.90), headache and cough (6.36 95% CI 2.83–14.33), and headache and fever (5.85 95% CI 1.85–18.47) (Table 3). 

### 3.3. Test Sensitivity and Specificity

Test sensitivity ranged from 0.311 ± 0.105 for OP iHealth to 0.957 ± 0.048 for saliva PCR. Test specificity was 1 (Table 4) for all AgRDT tests examined. Among individuals positive by any PCR, 68.0% were positive on all three PCR tests (Figure 2A) and among those positive for one or more PCR tests, 74.7% had at least one positive AgRDT. Individuals positive on any PCR test were positive on at least one nasal AgRDT and at least one OP AgRDT 34.7% of the time, positive on at least one nasal AgRDT but no OP AgRDTs 28.0% of the time, and positive on at least one OP AgRDT but no nasal AgRDTs 12.0% of the time (Figure 2B). The sensitivities of PCR tests were significantly higher than AgRDTs (all *p* < 0.0001). There were no significant differences in performance between BinaxNow and iHealth, for either nasal swabbing (*p* = 1.0) or OP swabbing (*p* = 0.14). With respect to the sample type: Nasal AgRDTs tests were more sensitive than OP AgRDTs (all *p* < 0.01). Saliva PCR was significantly more sensitive than OP PCR (*p* < 0.05), and trended towards greater sensitivity than NP PCR, though this did not reach significance (*p* = 0.11). NP and OP PCR testing were not significantly different (Figure 3). 

### 3.4. Sequenced Lineages

In total, 115 samples (including multiple samples from individual participants) were sequenced for phylogenetic classification using the Phylogenetic Assignment of Named Global Outbreak (PANGO, nomenclature system developed in 2020 based on phylogeny in order to track the most transmissible variants [19]) and Nextstrain (year-letter nomenclature designed to track pathogen evolution over time with open-source genomic data [20]) classification systems (Table 5). This yielded eight viral strains by Pango lineage and two Nextrain clades. Ba.1.1 and 21K (omicron) were the most common, respectively.

### 3.5. Cycle Threshold (Ct) Values

All PCR samples run on the Multiplex platform had available Ct values for the N gene, S gene, and ORF1ab. The analysis utilized only positive tests. Average Ct values were generally lowest in NP tests, followed by saliva tests, and highest in OP tests (Figure 4D). Ct values plotted against the number of positive antigen tests out of four demonstrated significantly lower Ct values associated with more positive antigen tests (Figure 5). 

## 4. Discussion

We compared seven test modalities for the diagnosis of COVID-19 in 325 participants. We found that PCR tests were the most sensitive, nasal AgRDTs were moderately sensitive, and OP AgRDTs were the least sensitive. Prior meta-analyses showed single AgRDTs to have a sensitivity of approximately 70%, and previous studies of BinaxNOW demonstrated a sensitivity of around 65% [5,6]. Nasal AgRDT test performance in our study did not differ significantly from these published results. PCR tests had a higher sensitivity than all AgRDTs, which is consistent with previous research on PCR test sensitivity [10]. 

The site of testing was an important factor influencing sensitivity. Saliva tests were significantly more sensitive than OP tests, but NP tests were not significantly different from saliva or OP tests. Interestingly, OP tests were associated with the highest PCR Ct values across gene targets, suggesting that this sample type yielded the lowest concentration of viral RNA material, which parallels the lower sensitivity of this method in our cohort. While NP samples produced somewhat lower Ct values than saliva samples, there was no significant difference in test performance between these sample types. Our results are consistent with prior research on PCR performance in NP and saliva samples spanning the delta and omicron waves showing higher viral load in NP samples but equivalent diagnostic sensitivity [14]. Saliva tests are more comfortable and affordable than NP tests [9,15], which combined with their equal sensitivity to NP tests would appear to make them a preferred test method. One drawback of salivary testing is the need for participants to abstain from eating or drinking for 30 min prior to sample collection which may create logistical challenges during mass testing. Even with this waiting period, food particles, tobacco products, and oral hygiene products pose a theoretical risk of contamination that remains under-studied. However, salivary testing can be performed by lay individuals, even as part of an unsupervised “drop-box” collection system with samples placed directly onto ice by patients. Improved comfort could also represent a substantial benefit when testing in the pediatric population. 

The use of oropharyngeal swabbing for AgRDTs is a non-standard technique, which in this study yielded lower sensitivity than nasal AgRDTs. Nevertheless, 9 out of 75 positive participants were positive on at least one OP AgRDT, but negative on both nasal AgRDTs, highlighting considerable test performance variability by swabbing site. The reason for this discrepancy is unknown, and could represent a variation in individual test kit performance, technical factors during swabbing and/or actual site-specific biological differences in the shedding of viral antigens. This observation suggests that sensitivity during point-of-care or home-use of AgRDTs could be optimized by the use of both an oral and nasal swab during testing. A study by Goodall et al. in 2022 examining the use of OP swabs and combined nasal/throat swabs during the omicron wave found that NP and OP swabs each had a sensitivity of around 0.645, but that combined nasal/throat swabs increased sensitivity to 0.887 relative to PCR [21]. Important differences between this study and our work include their use of asymptomatic participants, RT-PCR from residual viral media, Panbio brand tests, and sample self-collection. More research is needed to elucidate potential benefits of OP swabs in conjunction with nasal swabs and/or the utility of combined nasal/throat swabs. Our finding that participants with more positive rapid antigen tests have significantly lower average Ct values on their PCR tests is consistent with previous research demonstrating an inverse correlation with Ct values and AgRDT sensitivity [2,3,4]. This suggests that a higher viral load may be associated with AgRDT sensitivity.

While early predictive models had found loss of taste and smell, fatigue, cough, and anorexia to be highly correlated with a positive COVID-19 test [22], in our study the individual symptoms most predictive of a positive test were cough, chills, and fever. Chills and cough were the most significant predictive pair of a positive test. The loss of taste and smell were not commonly reported symptoms in our dataset. Fatigue was quite common but not predictive of a positive test. Other studies performed during the omicron wave reported patterns of symptomatology consistent with those observed here [23], underscoring the need for continued surveillance and clinical research during the evolution of a viral pandemic. Among 115 samples yielding viral genomic sequences, 82 were BA.1.1, and all were within the omicron family, suggesting most infections in our cohort were due to the omicron variant. A comparative assessment of symptoms and test performance by viral strain was therefore not possible with this study but may be an interesting area of further research. 

This study has several limitations. During data collection, some AgRDTs tests were run at outside ambient seasonal temperatures ranging from 36 °F to 60 °F. This complies with storage recommendations for both brands (35.6 °F and 86 °F). However, iHealth manufacturer instructions recommend running the assays between 65 and 86 °F, and slightly colder temperatures in our “real-world setting” may have affected test performance. Our data set only included one asymptomatic positive individual, who was positive on all three PCR tests but negative on all AgRDTs. A 2023 meta-analysis showed a decreased sensitivity at 42.6% when AgRDTs were used as screening tools in the general population [4]. Therefore, AgRDTs may be most valuable for maximizing the early detection of COVID-19 in symptomatic individuals but should not be relied on exclusively for all testing purposes. Additionally, a 2023 multicenter randomized trial with a large proportion of asymptomatic individuals found saliva sensitivity to be significantly lower than NP sensitivity [24]. With only one asymptomatic positive individual, our results may not be generalizable to a largely asymptomatic screening population. In this study, participants were recruited by sequential convenience sampling. Since respiratory illnesses other than COVID-19 were not assessed, we were unable to assess cross-reactivity between COVID-19 and other pathogens such as RSV and influenza.

## 5. Conclusions

PCR COVID-19 tests are the most sensitive and should remain the gold standard for COVID-19 detection. Saliva PCR is more sensitive than OP PCR, and offers financial, operational, and patient comfort advantages compared to NP PCR. AgRDTs are less sensitive than PCR tests. AgRDTs conducted in the standard nasal manner are more sensitive than AgRDTs conducted with oropharyngeal samples. Differences in both test sensitivity and COVID-19 symptom presentation between our data and older, pre-omicron data reinforce the need for the continued assessment of tests and risk stratification tools, both for currently prevalent respiratory viruses and future viral pandemics. 

## Figures and Tables

**Figure 1 diagnostics-14-00986-f001:**
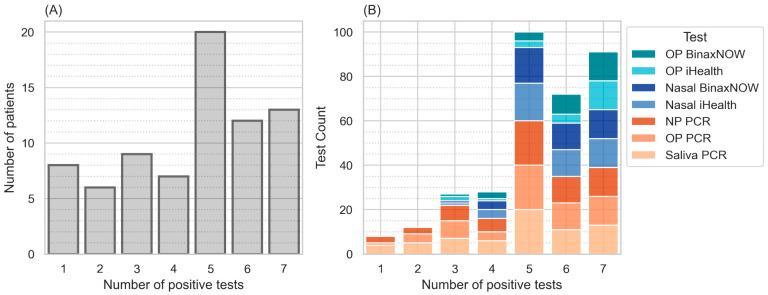
(**A**): Subjects by number of positive tests. This shows the distribution of participants by number of positive tests (maximum possible = 7). (**B**): Distribution of test methods by number of positive tests. For patients with only one or two positive tests out of seven, only PCR tests were positive. In subjects with five or more positive tests out of seven, all PCR tests were positive.

**Figure 2 diagnostics-14-00986-f002:**
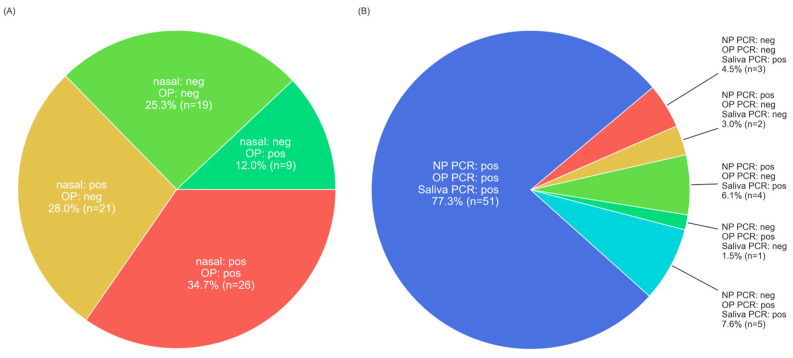
Test positivity pattern breakdown by sample type in COVID-positive participants. (**A**): Antigen test positivity by test site given a positive PCR: Among 75 patients were positive on at least one PCR test, 25.3% were negative on all nasal and OP AgRDTs; 12.0% were positive on at least one OP AgRDT but negative on both nasal AgRDTs; 28.0% were positive on at least one nasal AgRDT but negative on both OP AgRDTs; and 34.7% were positive on both nasal and OP AgRDTs. (**B**): PCR test positivity by test site in COVID-19 positive individuals: Most participants with at least one positive PCR test (77%%) were positive on all three PCR tests (NP, OP, and saliva). The remaining percentages are as shown.

**Figure 3 diagnostics-14-00986-f003:**
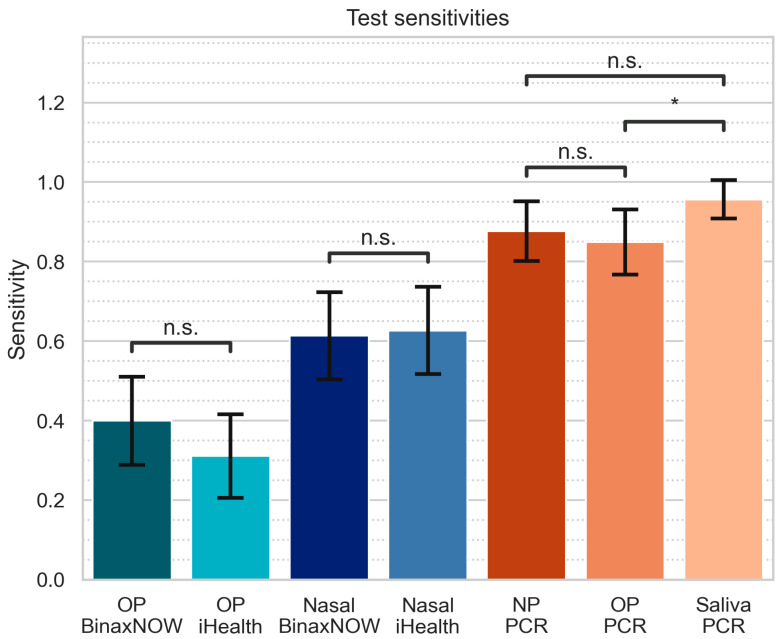
Sensitivities of all tests. “n.s.” denotes no significant difference between tests. “*” denotes significant difference with *p* < 0.05. All other pairwise comparisons not marked are significantly different with *p* < 0.01. Error bars represent one standard deviation.

**Figure 4 diagnostics-14-00986-f004:**
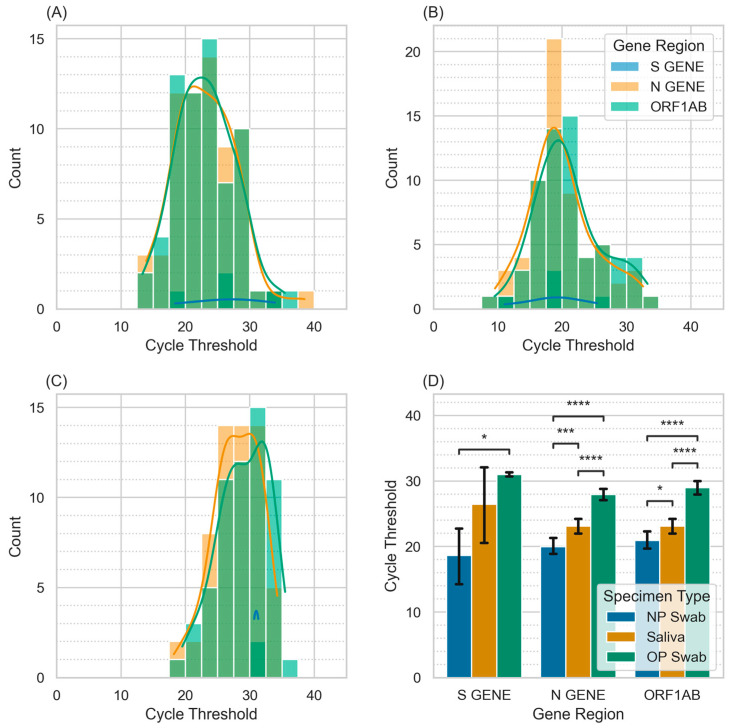
Distribution of PCR cycle thresholds by sample type and gene region. (**A**–**C**): Distribution of sample Ct values by specimen type. (**A**): Saliva sampling. (**B**): Nasopharyngeal sampling (NP). (**C**): Oropharyngeal sampling (OP). (**D**): Average cycle thresholds by specimen type and gene region. *: *p* < 0.05; ***: *p* < 0.001; ****: *p* < 0.0001.

**Figure 5 diagnostics-14-00986-f005:**
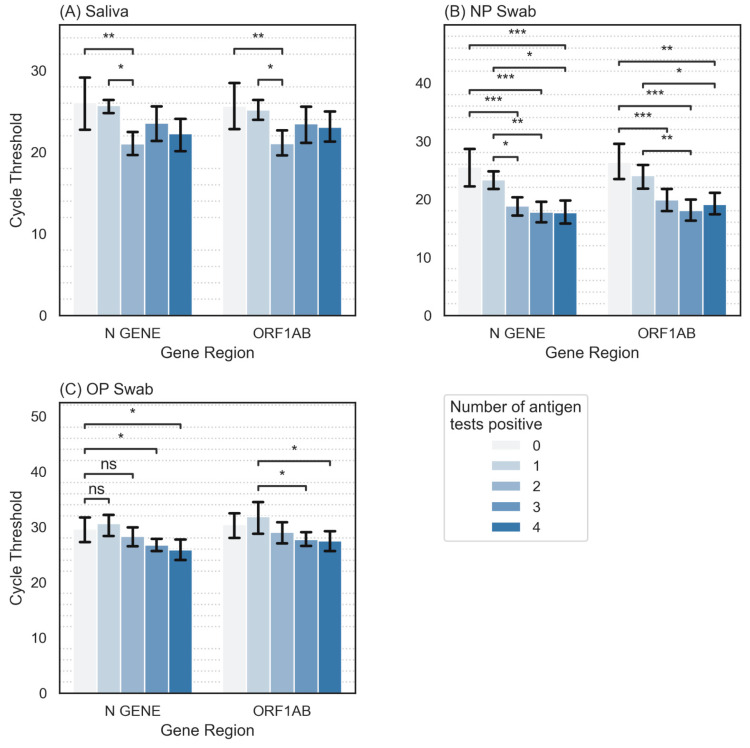
Distribution of mean Ct values for N gene and ORF1ab across number of positive rapid antigen tests (0–4). (**A**): Saliva sampling. (**B**): Nasopharyngeal sampling (NP). (**C**): Oropharyngeal sampling (OP). *: *p* < 0.05; **: *p* < 0.01; ***: *p* < 0.001.

**Table 1 diagnostics-14-00986-t001:** Frequency of reported symptoms. Symptoms reported by all participants (left columns), participants positive on any COVID-19 test (middle columns), and participants positive on no COVID-19 tests (right columns). Total number of participants and proportion out of participant group (all, positive, or negative) reporting each symptom are shown.

Symptom	All Participants		Positive Participants		Negative Participants	
	Number Reporting	Proportion Reporting	Number Reporting	Proportion Reporting	Number Reporting	Proportion Reporting
Sore throat	165	0.5077	46	0.6133	119	0.4760
Cough	87	0.2677	43	0.5733	44	0.1760
Headache	82	0.2523	24	0.3200	58	0.2320
Congestion	76	0.2338	18	0.2400	58	0.2320
Fatigue	60	0.1846	12	0.1600	48	0.1920
Myalgias	60	0.1846	23	0.3067	37	0.1480
Rhinorrhea	43	0.1323	8	0.1067	35	0.1400
Fever	36	0.1108	18	0.2400	18	0.0720
Chills	21	0.0646	10	0.1333	11	0.0440
GI distress	14	0.0431	0	0	14	0.0560
Nausea	12	0.0369	0	0	12	0.0480
Dyspnea	6	0.0185	3	0.0400	3	0.0120
Chest tightness	5	0.0154	2	0.0267	3	0.0120
Dizziness	5	0.0154	2	0.0267	3	0.0120
Sneezing	4	0.0123	1	0.0133	3	0.0120
Loss of smell	4	0.0123	2	0.0267	2	0.0080
Sinus pressure	3	0.0092	1	0.0133	2	0.0080
Ear pain	3	0.0092	2	0.0267	1	0.0040
Hot flashes	2	0.0062	0	0	2	0.0080
Neck stiffness	1	0.0031	0	0	1	0.0040
Loss of taste	1	0.0031	0	0	1	0.0040
Night sweats	1	0.0031	1	0.0133	0	0
Loss of appetite	1	0.0031	0	0	1	0.0040
Insomnia	1	0.0031	0	0	1	0.0040

**Table 2 diagnostics-14-00986-t002:** Odds ratio of receiving any positive test with the listed symptom present compared to the odds of not receiving a positive test with the listed symptom present. NA indicates confidence interval is not applicable due to odds ratio of zero. GI distress combines vomiting and diarrhea.

Symptoms	Odds Ratio	95% CI
Sore throat	1.75	(1.03–2.96)
Cough	6.29	(3.59–11.03)
Headache	1.56	(0.88–2.75)
Congestion	1.09	(0.60–2.01)
Fatigue	0.80	(0.40–1.60)
Myalgias	2.47	(1.34–4.54)
Rhinorrhea	0.73	(0.32–1.66)
Fever	4.07	(1.99–8.32)
Chills	3.34	(1.36–8.22)
Nausea	0.00	NA

**Table 3 diagnostics-14-00986-t003:** Odds ratios (OR) for receiving at least one positive test given the presence of two symptoms compared to the odds of receiving a positive test with neither test present. A 95% confidence interval is present below each OR. NA indicates confidence interval is not applicable due to odds ratio of zero.

	Sore Throat	Cough	Headache	Congestion	Fatigue	Body Aches	Rhinorrhea	Fever
cough	5.06							
	(2.71–9.48)							
headache	1.43	6.37						
	(0.71–2.89)	(2.83–14.33)						
congestion	0.95	4.21	1.87					
	(0.44–2.01)	(1.93–9.21)	(0.76–4.60)					
fatigue	0.75	2.30	1.12	0.61				
	(0.30–1.90)	(0.95–5.55)	(0.35–3.57)	(0.17–2.15)				
body aches	3.04	4.56	2.22	0.70	0.70			
	(1.46–6.33)	(1.95–10.66)	(1.00–4.93)	(0.20–2.51)	(0.20–2.51)			
rhinorrhea	0.50	0.95	0.36	0.00	1.69	0.95		
	(0.17–1.51)	(0.30–2.98)	(0.05–2.90)	(NA)	(0.41–6.95)	(0.19–4.68)		
fever	4.60	5.76	5.85	2.63	0.66	2.63	0.000	
	(1.83–11.58)	(2.26–14.70)	(1.853–18.47)	(0.88–7.84)	(0.08–5.76)	(0.88–7.84)	(NA)	
chills	2.91	29.73	3.47	2.04	0.55	2.76	3.37	4.64
	(0.86–9.80)	(3.65–241.90)	(0.85–14.20)	(0.88–7.84)	(0.07–4.64)	(0.72–10.55)	(0.21–54.46)	(1.01–21.21)

**Table 4 diagnostics-14-00986-t004:** Sensitivity and specificity of all tests measured against any positive PCR as gold standard.

Test	Sensitivity	Specificity
OP BinaxNOW	0.400 ± 0.111	1.000 ± 0.000
OP iHealth	0.311 ± 0.105	1.000 ± 0.000
Nasal BinaxNOW	0.613 ± 0.110	1.000 ± 0.000
Nasal iHealth	0.627 ± 0.109	1.000 ± 0.000
NP PCR	0.877 ± 0.075	1.000 ± 0.000
OP PCR	0.849 ± 0.082	1.000 ± 0.000
Saliva PCR	0.957 ± 0.048	1.000 ± 0.000

**Table 5 diagnostics-14-00986-t005:** Number of samples in each Pango lineage and Nextstrain clade.

Pango Lineage	Number of Samples	Nextstrain Clade	Number of Samples
BA.1.1	82	21K (omicron)	111
BA.1.15	9	21L (omicron)	4
BA.1.20	8		
BA.1.1.18	4		
BA.2	4		
BA.1.17	4		
BA.1	2		
BA.1.17.2	2		

## Data Availability

All deidentified data will be made publicly available on request by writing to tornberg@ohsu.edu.

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
