# Peer review of "Comparative Performance of COVID-19 Test Methods in Healthcare Workers during the Omicron Wave"

_diagnostics, 2024, doi:10.3390/diagnostics14100986_

Round 1

Reviewer 1 Report

Comments and Suggestions for Authors

This is a very informative and well written paper. I have only a few minor language edits to suggest and comments:

1. Line 30 and throughout: It think the abbreviation NAT is more commonly used for nucleic acid testing, but no change is required if you prefer NAAT.

2. For the uninitiated (such as me) some expansion of the Section 3.1.4 might be useful to help readers understand what is meant by "Pango lineage" and "Nextstrain Clade".

3. Figure 1 legend: I think the second sentence should be "This shows the distribution of participants by number of positive tests......"

4. Figures 3 and 4: the pie figures are a little hard to read. Can the size be increased in the article?

5. Lines 236-238: I think you meant to say "Our data set only included one asymptomatic positive individual, who was positive on all three PCR tests but negative on all AgRDTs."

Comments on the Quality of English Language

very nicely written

Author Response

Thank you so much for your review of our paper. 

  1. Line 30 and throughout: It think the abbreviation NAT is more commonly used for nucleic acid testing, but no change is required if you prefer NAAT.

Response: Thank you for this comment, NAAT is commonly used by the CDC and NIH which is why we utilized that abbreviation over NAT.

2. For the uninitiated (such as me) some expansion of the Section 3.1.4 might be useful to help readers understand what is meant by "Pango lineage" and "Nextstrain Clade".

Response: 

An explanation of Nextrain and Pango was added along with appropriate citations in lines 153-158: 

"One hundred and fifteen samples (including multiple samples from individual participants) were sequenced for phylogenetic classification using the Phylogenetic Assignment of Named Global Outbreak (PANGO, nomenclature system developed in 2020 based on phylogeny in order to track the most transmissable variants [19]) and Nextstrain (year-letter nomenclature designed to track pathogen evolution over time with open-source genomic data [20]) classification systems (Table 5) This yielded eight viral strains by Pango lineage and two Nextrain clades. Ba.1.1 and 21K (Omicron) were the most common, respectively."

3. Figure 1 legend: I think the second sentence should be "This shows the distribution of participants by number of positive tests......"

Response: Change was made as recommended. 

4. Figures 3 and 4: the pie figures are a little hard to read. Can the size be increased in the article?

Response: Size of text in pie charts was increased.

5. Lines 236-238: I think you meant to say "Our data set only included one asymptomatic positive individual, who was positive on all three PCR tests but negative on all AgRDTs."

Response: Edit made per your recommendation now in lines 236-238. 

Reviewer 2 Report

Comments and Suggestions for Authors

 Dear Editor

In the presented article, several diagnostic tests have been compared,
and some points should be taken into consideration and corrected
.

 In the abstract, the conclusion is not provided along with suggestions for the reader, only the statistical test results are presented.

In the methods section, the following points should be addressed:

  • The statistical population could have been better selected and grouped for analysis.
  • The definite negative group should consist of volunteers without vaccination history or disease symptoms.
  • Including a group with other respiratory diseases would help assess specificity tests and probably cross reactions.

In the results, Figure 5 only highlights two groups with p<0.05 significance, while the explanation mentions other groups with p<0.01 significance. It is recommended to redesign the figure to show differences based on p<0.01 and discuss these findings in the discussion section.

Good luck

Comments on the Quality of English Language

Minor editing of English language required

Author Response

Dear Reviewer 2, 

Thank you for your thoughtful review of our paper. We have made changes and are very grateful for your input on this work. Here is a point-by-point response: 

  1. In the abstract, the conclusion is not provided along with suggestions for the reader, only the statistical test results are presented.

Response: We have added "Saliva-based PCR testing is a viable alternative to traditional swab-based PCR testing for the diagnosis of COVID-19." in lines 22-23 following, "PCR remains the most sensitive testing modality for COVID-19, with saliva PCR being significantly more sensitive than oropharyngeal PCR and equivalent to nasopharyngeal PCR. Nasal AgRDTs are less sensitive than PCR but have benefits in convenience and accessibility."

2. In the methods section, the following points should be addressed:

  • The statistical population could have been better selected and grouped for analysis.
  • The definite negative group should consist of volunteers without vaccination history or disease symptoms.
  • Including a group with other respiratory diseases would help assess specificity tests and probably cross reactions.

Response: 

Reply: We thank the reviewer for this important comment. We acknowledge that a robust study design alternative could have involved selection of participants into specific groups, and inclusion of unvaccinated individuals along with persons experiencing respiratory infections other than COVID-19. However, due to the specific setting (occupational health intake), the high prevalence of COVID-19, and the universal vaccination status of OHSU employees (approaching 100% at the time of study), the extreme challenge of recruiting participants willing to undergo 7 separate COVID-19 tests, and the relatively low prevalence of other respiratory illnesses due to the high uptake of mask usage, we employed an unselected sequential convenience sample approach to participant recruitment. While our design is not optimized for evaluation of cross-reactivity between COVID and other respiratory diseases (test specificity), it did offer the opportunity to robustly test the specific hypotheses presented here related to test sensitivity. We note that specificity of the various COVID-19 tests is generally considered to be very high and may be less problematic than sensitivity, but is still worthy of study in an appropriately designed study. We attempt to address this in the limitations section of the manuscript.

Changes made (Line 248): In this study, participants were recruited by sequential convenience sampling. Because respiratory illnesses other than COVID-19 were not assessed, we were not able to assess cross-reactivity between COVID-19 and other pathogens such as RSV and influenza.

3. In the results, Figure 5 only highlights two groups with p<0.05 significance, while the explanation mentions other groups with p<0.01 significance. It is recommended to redesign the figure to show differences based on p<0.01 and discuss these findings in the discussion section.

Response: We had tried to visually demonstrate all differences based on p<0.01 previously, but it made the figure quite visually busy, so opted to list this explanation in writing. If you would like, we can send you this other image to see if you would prefer it for the paper. We have added the word "pairwise" before comparisons for increased clarity. 

Round 2

Reviewer 2 Report

Comments and Suggestions for Authors

-

Comments on the Quality of English Language

Minor editing of English language required

Author Response

Dear Reviewer, 

Thank you again for your feedback on our paper. We have reviewed the manuscript independently for grammar and spelling and have made minor grammatical edits throughout. Let us know if you suggest any other edits.

Sincerely, 

Emma Tornberg